# Dataset of Specific Total Embodied Energy and Specific Total Weight of 40 Buildings from the Last Four Decades in the Andean Region of Ecuador

Jefferson Torres-Quezada [1,*] and Tatiana Sánchez-Quezada [2]

1   City, Environment and Technology Research Group, Catholic University of Cuenca, Cuenca 010102, Ecuador
2   Technic University of Machala, Machala 070201, Ecuador
*   Correspondence: jefferson.torres@ucacue.edu.ec

**Abstract:** This article presents the Specific Total Embodied Energy (STEE) and Specific Total Weight (STW) of 40 Andean residential buildings in Ecuador, from 1980 to 2020. Firstly, the BoM of ten buildings of every decade was obtained through field work carried out in three urban sectors of this city. Secondly, the specific embodied energy and specific weight of every material found in the 40 samples were obtained by bibliography. Finally, the calculation of each building was divided into three components: Structure, Envelope and Finishes. The analyzed data show a detailed collection of different materials and construction typologies used in these four decades, and the impact on their embodied energy and their weight. Moreover, this article gives a Specific Embodied Energy and Specific Weight database of 25 materials that are extensively used in Andean regions. The results show several changes in reference to the insertion of new material, but also regarding the adoption of new architectonic models. The most important changes, in the analyzed period, have been the use of concrete and metal in the structure instead of wood, the increase in the glass surface in the envelope, and the replacement of wood by particleboard on the finishes. In conclusion, the STEE of the entire building has experienced an increase of 2.19 times in the last four decades. The STW value has also increased, but to a lesser extent (1.36 times).

**Dataset:** https://data.mendeley.com/datasets/z2pf5hzsss.

**Dataset License:** CC-BY-4.0.

**Keywords:** embodied energy; weight of building; structure; envelope; finishes

## 1. Summary

The construction sector consumes approximately 48% of the world's energy [1]. This energy is mainly generated from fossil fuels, which contributes significantly to $CO_2$ emissions. This energy is consumed, on the one hand, in an operative lifetime through the use of lighting, equipment, heating and cooling. On the other hand, buildings also consume energy during the construction stage due to the extraction, transportation and manufacturing of the materials used [2]. These two energies are called Operational Energy and Embodied Energy, respectively, and they comprise the overall building life cycle energy [3].

In recent decades, most sustainable strategies and regulations have focused on reducing the operational energy of buildings [4] principally due to the use of active climatization systems [5]. With this purpose, the use of insulation, double glazed windows and innovate materials has taken an increasing role in residential buildings [6]. Moreover, new technologies have been included to increase construction productivity [7]. However, these changes tend to increase the Embodied Energy of the buildings [8]. Additionally, these changes have caused traditional construction to experience a significant decline in recent decades, especially in Latin American countries [9] such as Ecuador.

In the four natural regions of Ecuador (Coast, Amazonian, Insular and especially the Andean region), vernacular practices and materials have been replaced by new construction systems and industrial materials. In addition, recent construction practices have adopted imported architectural models, which have generated a loss of diversity and identity in the local architecture [10,11]. Consequently, concrete, pumice block and fiber cement have replaced the use of wood beams, load-bearing mud walls and artisanal tiles in residential Andean buildings.

Worldwide, several studies have focused on analyzing the Total Embodied Energy per unit area (STEE: $GJ/m^2$). The most useful boundary conditions in these studies are Cradle-to-Gate (CG) and Cradle-to-Site [12]. In India, a building with a load-bearing stone masonry wall construction system has an STEE of around 1.20 $GJ/m^2$, while a building with a reinforced concrete structure with burnt brick masonry is around 4.00 $GJ/m^2$ [6]. In Brazil, in typical constructions made of reinforced concrete and structural masonry, the STEE value is around 1.50 $GJ/m^2$ [13]. In Dutch dwellings with an insulated envelope, the STEE goes from 3.0 to 6.4 $GJ/m^2$ [14]. Four-story residential buildings in Slovenia, characterized by high insultation standards and timber structure, have a value of around 5.90 $GJ/m^2$ [15].

Additionally, several studies have determined that the component of the building that has the greatest impact on the demand for Embodied Energy is the Structure. Su and Zhang [16] evaluated three residential buildings (6, 11 and 18 stories) built with metal structure in China. This study determines that the Structure components represent between 53% and 59% (3.33–3.69 $GJ/m^2$) of the entire STEE value of the building. The STEE value in these buildings is influenced to a greater extent by the EE value of the metal than by its weight, which was between 1.3 $kg/m^2$ and 1.5 $kg/m^2$. The opposite happens with concrete structures, where the weight is the determining feature. Due to the importance of the structure, strategies are evaluated to reduce the impact of this component, such as the use of recycled aggregates [17,18], or the use of wood [19,20]. However, these strategies have weaknesses in their resistance and rigidity that result in the addition of other materials, which implies an increase in their demand for embodied energy [15,21].

All these studies give a perspective on the high impact of industrialized construction systems on the demand for Embodied Energy in buildings in different countries (Asia and Europe). In addition, they point out the importance of applying strategies focused on the Structure component. However, there are few studies that consider the contribution that the Envelope and Finish components can generate. Most of these strategies focus on the use of recycled materials or materials of more natural origin without considering the architectural decisions that define the building or the changes experienced over time.

Latin American countries such as Ecuador, which have begun to experience the shift from vernacular to industrialized building systems in recent decades, have negligible scientific production on this topic, mainly due to the lack of a database of EE indexes for each material [22,23].

Based on above, the database presented in this article addresses the existing information gap on the estimated value of total building Embodied Energy demand in the Andean regions of Ecuador. The research significance of this article is based on a vision of the sample that covers a broader field based on the architectural and constructive changes that construction undergoes over time. Therefore, this article presents detailed data (constructive and energetic) on these changes in the last four decades, focused not only on the Embodied Energy of the Structure component, but also on its Envelope and Finishes.

Consequently, this article presents an extensive database, on which the related article [24] is based, regarding:

- The specific embodied energy of 25 materials widely used in Andean regions;
- The Bill of Materials (BoM) of ten buildings from every decade, with different construction systems such as brick walls, concrete, metal or wood;
- The STEE and STW value of the Structure, Envelope and Finishes component.

These data can be useful for researchers, engineers and architects, in further studies, future constructions and design decisions with a sustainable approach.

- Due to the lack of embodied energy information on construction materials in Latin American countries [22], a variety of projects can come out of the primary dataset presented in this study;
- The energy impact of different construction systems and their respective components such as Structure, Envelope and Finishes are presented;
- The Total Specific Weight of each material and component included in this article can be used in other analyses, not only those focused on the embodied energy, but also on operational energy.

## 2. Data Description

The data presented in this article focus on two parameters: the Specific Total Embodied Energy (STEE) and the Specific Total Weight (STW) obtained from forty residential buildings in Cuenca, Ecuador. STEE and STW refer to the Total Embodied Energy and Total Weight, respectively, of the entire building, component or material divided by its total floor area ($MJ/m^2$). The data were obtained from four different periods named P1 (1980 to 1990), P2 (1991 to 2000), P3 (2001 to 2010) and P4 (2011 to 2020). Ten samples were calculated from every period.

The collected data have been organized into three components, Structure, Envelope and Finishes, for every building analyzed. Table 1 summarizes the general information related to the databases.

**Table 1.** Database information.

| | |
|---|---|
| **Type of data** | Four databases in spreadsheet format (json format)01_EE_W_index 02_BoM_STEE_STW_calculations 03_STEE_STW_materialscomponents 04_STEE_STW_entire_building |
| **Data source location** | The field work was carried out in 40 residential buildings located in Cuenca, Azuay, Ecuador (Andean region). |
| **Specific subject area** | The research addresses the area of architecture design from the perspective of the life cycle of materials, with a specific focus on their embodied energy. |
| **Climate** | Annual average temperature: 15.50 °C Annual oscillation: 2 °C Daily oscillation: 10 °C |
| **Related research article** | The evolution of embodied energy in Andean residential buildings. Methodology applied to Cuenca, Ecuador. Energy and Buildings, 2022 [24] |

The data presented in this article are organized in four parts: (1) the Specific Embodied energy and Specific Weight index of every material found in the forty buildings analyzed, (2) the Bill of Materials (BoM) of each building, with the calculations of STEE and STW, (3) the STEE and STW organized by material, element and component and (4) a summary table of STEE and STW.

*2.1. Samples Description*

Two construction system types from each period were recognized, depending mainly on their Structure and Envelope.

In the first period, the first type had a continuous cyclopean concrete foundation. The envelope had walls made of 30-cm-thick solid brick, and the roof was made of wood, fiber cement and a double layer of artisanal tile. The second type had an isolated reinforced concrete foundation with concrete columns and 15-cm-thick solid brick walls. In this period, type 1 was the most used, with six of the ten samples (houses 1, 2, 3, 6, 7 and 10), while houses 4, 5, 8, and 9 corresponded to Type 2.

In the second period, two construction system types were recognized with the same characteristics shown in period 1. However, type 2 was the predominant typology with houses 2, 4, 5, 6, 7, 8, 9 and 10. Only houses 1 and 3 corresponded to type 1.

In the third period, type 1, with nine of the ten samples, had a reinforced concrete isolated foundation plus a continuous cyclopean concrete foundation. Type 2 had only an isolated foundation made of concrete. These two typologies are very similar in reference to the envelope. They are characterized by 15-cm-thick solid brick walls in the first floor. In this period, hollow brick appeared for the walls of the second floor.

In the fourth period, type 1 had a structure made of metal while type 2 used reinforced concrete. In both typologies, the foundation was made of reinforced concrete plus cyclopean concrete. One of the most important changes compared to the other periods is the use of horizontal concrete slabs for roofs. In period 4, houses 1, 6, 7 and 9 corresponded to type 1, while houses 2, 3, 4, 5, 8 and 10 corresponded to Type 2.

Detailed information of these samples is shown in Database 1 of this article. Furthermore, the figures from related article [24] show the construction systems described in each period graphically.

*2.2. Database 01: Specific Embodied Energy (SEE) and Specific Weight (SW) of 25 Materials of Andean Construction Systems*

According to the collected information, 25 materials were found in the forty buildings analyzed. Since this article addresses the evolution of construction systems in four periods, it was necessary to consider the improvements of materials in the Ecuadorian industry. These changes were consolidated only in the fourth period. The most important changes had an impact on concrete, steel, stainless steel and aluminum. Thus, two SEE values of these four materials have been specified in the supplementary file named "01_EE_W_index". The first value does not consider the improvements industry, and the second value, which is marked with "*", considers these improvements. The supplementary file is organized as follows:

- Materials (column A);
- Specific Embodied Energy_SEE (Column B);
- SEE source (column C);
- SEE country (column D);
- SEE boundary condition (column E);
- Specific Weight_SW (column F);
- Weight source (column G).

The SEE and SW of these materials (25) were obtained by bibliography. In the case of SW, the information corresponds only to local information achieved from Ecuadorian industries. However, most of the SEE values were acquired from international databases since there was a lack of information in Ecuador, as well as in several South American countries. Numerous SEE values of each material were collected. However, the values chosen and shown in this article correspond to those that maintained manufacturing, extraction and transportation processes that were most similar to the manufacturing processes carried out in Ecuador. This selection process is explained in detail in the Methods section.

The characteristics of these twenty-five materials, as well as the four derivations due to industry improvements, are explained below.

All the materials detailed below are manufactured nationally. Although there are options for imported materials, these have not been considered in this study.

The first material is concrete, with a resistance of 210 kg/cm$^2$ (21 MPa) and a dosage of 1:2:3 (cement/sand/gravel). The manufacture of cement and its raw materials and aggregates come from local quarries (50 km).

The concrete with industrial improvements, used only in period 4, was defined based on the sustainability regulations applied in the industries from 2008 and consolidated in the last decade [25].

Like the concrete, the materials used for the cement mortar came from quarries in the city (50 km). This material is manufactured on site. The proportion used was 1:4 (cement/sand).

Lightweight pumice blocks were manufactured in small-scale, semi-industrialized factories. This process begins with the transportation of raw material from quarries close to the factories (35 km). The raw material corresponds to a percentage of clinker plus pumice stone. This raw material is stored abroad. After that, we proceeded to mix the raw material with water and additive through processes carried out with human energy. This mixture was put into an electric press machine, where each one of the pieces was shaped, and finally they were dried in the open air [26].

The structural stone was obtained from quarries located on the outskirts of the city (50–70 km) and was transported directly to the construction site.

The artisanal solid brick is a type of brick that has been widely used since the 19th century in the Andean region of Ecuador. The manufacture of this material occurs in artisan factories that are located in the north of the city in the parishes of Sinincay, Sayausí and San Sebastián (12 km). The production of this material begins with the extraction of the raw material on site, without using any type of machinery. This material is classified and usually goes through the mixing process (earth plus water) with animal power. This mixture is placed in molds and then cooked in ovens with firewood of natural origin [27]. The production process has changed in a minimal proportion throughout four decades, and therefore continues to be artisanal [28].

The artisanal hollow brick has a very similar process to the previous material. The stage in which it varies is in the forming process, where mechanical equipment is used to shape the block, although it is operated with human force [27]. Therefore, the EE of this material does not change compared to the artisanal solid brick.

Artisanal tile is manufactured in a semi-industrial factory with a similar process as the solid or hollow brick. The most significant change is in the mixing stage, where extrusion and mixing machines that work with diesel fuel are used [29].

The vitrified tile is made up of a ceramic tile plus an additional layer of enamel paint. Therefore, the manufacturing process differs from the artisanal tile due to its longer firing time, a glazing process and a second firing process to seal the final coating. The industries that make both tiles are located on the outskirts of the city (12 km).

The manufacture of vitrified ceramics uses the same process that is used for the production of bricks or tiles. However, in this case, the process is fully industrialized in factories located on the outskirts of the city. The raw material goes through a process of selection, shredding, mixing and grinding. This raw material is stored in silos to later go through the process of kneading, molding, cutting and drying. Its final phase is firing in an oven, which reduces the humidity of the pieces to improve the performance of the material. To vitrify the ceramic, an enameling process is carried out made from varnish and a second firing of the material is carried out. All these processes use electrical energy and energy from fossil fuels [30].

Porcelain tile, like ceramic, has a fully industrialized production process. In this case, the selection process of the raw material and the crushing process are more exhaustive than in the manufacture of ceramics. In addition, the biggest difference occurs in the firing process, since porcelain tiles require a higher temperature and longer firing time [30].

For the manufacture of primary aluminum, most factories import the raw material in aluminum ingots, since Ecuador is not a producer of this material. These ingots go through an extrusion process to convert the material into profiles [30].

On the other hand, in the last decade most companies, in addition to importing the raw material, used recycled aluminum to obtain the ingots and thus manufacture the recycled aluminum (33%) [30]. The aluminum production factories are located 200 km from the study city. Finally, even when the raw material has been imported, this condition has not been considered for the EE values. Given the large number of imported tons of aluminum, the increase in the value of EE due to this item will be minimal.

The virgin steel production process is carried out in fully industrialized factories where electrical energy and mainly energy from fossil fuels are used. Until 2005, all the processes used virgin material from national and imported extraction [30].

On the other hand, as of 2010, the main companies that produce steel materials carry out processes of recycling and reusing scrap to produce recycled steel (40%) [31].

As for stainless steel, the production process is similar to steel. However, elements such as chromium, carbon, nickel and molybdenum are added to this material, which makes it possible to give the material a high purity and great resistance to rust and corrosion [32]. After obtaining the desired composition, it is passed to a hot lamination and a cold lamination that gives the final shape to the material requirement. Like steel, starting in 2010, factories began to use recycled material for the manufacture of raw materials.

The process for the manufacture of galvanized steel begins with obtaining the steel to be cold or hot rolled. Finally, this material goes through a galvanizing process using a galvalume coating (aluminum + zinc) to protect the steel from corrosion [33]. Again, the use of recycled materials began in 2010, and from this year recycled stainless steel was used. It is necessary to mention that the factory for these six recycled materials is located 200 km from the study city.

Glass production is carried out in industrialized factories located 150 km from the study city. The raw material is obtained from national quarries. It is necessary to point out that, for the production of building glass, 0% recycled materials are used due to the high difficulty of their collection and transportation [34].

The fiber cement sheets used in roofs are made up of cement, silica, cellulose fibers and additives. This material is manufactured in industrialized companies located on the periphery of the study city. All raw material is extracted from local quarries [35].

Vegetable fiber artisanal plaster panels which are used on the ceilings of homes are produced by hand in small workshops within the city. This product is made with raw material obtained from nearby places. The materials used are plaster and vegetable fibers.

Plaster is a material used to fill walls, made from lime and mixed with acrylic resin. This material is manufactured by local companies [36].

Plasterboard is an industrialized material produced by local factories. The raw material comes from national quarries 100 km away [37].

The plasterboard Structure is made with non-structural galvanized steel. The manufacture of this material is carried out in factories of the study city [37].

The plastic paint is manufactured in local factories, while the raw materials are obtained from national factories located in another region (300 km) [38].

Laminated wooden and veneer particleboard flooring is mainly imported from Asia or Europe. However, there are some national companies that manufacture this material. One of them is located in the study city. These materials are made from shredded wood, with the use of paints and synthetic resins for surface finishes. These resins are mostly imported. On the other hand, the production of wood is totally national [39].

Timber is used in different construction spaces, such as in the structure, floor coverings and masonry elements. The wood is obtained through the felling of trees in forests planted for this purpose located 50 km from the study city. The felled trees are taken to the sawmill to be cut and dried in the open air. In the country, drying is not carried out using ovens [40].

*2.3. Database 02: BoM, STEE and STW Calculations of the Ten Buildings of P1, P2, P3 and P4 + P4\**

This database corresponds to the BoM and the calculations of STEE and STW of the forty buildings. The forty buildings from each period are shown in the file named "02_BoMSTEESTW_calculations". The data shown in these files are organized into three components: Structure, Envelope and Finishes, which are classified by elements. The Structure component has seven elements: Foundations, Columns, Beams, First Floor, Second Floor, Roof and Stairs; the Envelope component has three: Roof, Walls and Windows; and the Finishes component has six: Floor, Doors, Wall finish, Handrails, Ceilings and Fitted Furniture. The construction system and all materials of each element have been specified. Furthermore, the total floor area and the year of construction of the building are shown in cells R2 and S2, respectively. Therefore, the data shown in this file are organized as follows:

- Period/House (column A);
- Component (column B);
- Element (column C);
- Description system (column D);
- Unit (column E);
- Quantity (column F);
- Materials (column G);
- Unit (column H);
- Quantity (column I);
- Specific Weight (column J);
- TW (obtained by the multiplication of I × J_ column K);
- Specific Embodied Energy by material (column L);
- TEE by material (obtained by the multiplication of K × L_ column M);
- TEE by element (sum of all the TEE values of all the materials that make up the element_ column N);
- TEE by component (sum of all the TEE values of all the materials that make up the component_ column O);
- STEE (obtained by the division of O/R_ column P);
- STW (obtained by the division of K/R_ column Q).

This database presents the real tendencies that the construction system has experienced in the last four decades. The data of the first four periods (P1–P4) shown in this file consider the industry improvements. Nevertheless, it was deemed necessary to share the STEE values of the last period that do not consider the improvements in the industry. Therefore, these data, named P4\*, are displayed in this same file.

*2.4. Database 03: STEE and STW Organized by Material, Element and Component*

This database shows the STEE and STW values organized by materials of every component. The data are shown in the file named "03_STEE_STW_materials&components". The file contains the information of the ten houses in each period for both STEE and STW values. Three values are shown for each house in the STEE (the Total Embodied Energy by material, Specific Total Embodied Energy by material and the Percentage by material) and STW (the Total Weight by material, Specific Total Weight by material and the Percentage by material) sections. In addition, each section shows the average of the ten houses by material. This information is organized as follows:

- Period (column A);
- Components (column B);
- Element (Column C);
- Materials (Column D);
- TEE by material (Columns E, H, K, N, Q, T, W, Z, AC and AF);
- STEE by material (obtained by the division of TEE/total floor area_ Columns F, I, L, O, R, U, X, AA, AD and AG);

- Percentage by material (obtained by the division of STEE by material/STEE of entire component_ Columns G, J, M, P, S, V, Y, AB, AE and AH);
- Average STEE by material (obtained by the average of the 10 buildings' STEE_ Column AI);
- TW by material (Columns AJ, AM, AP, AS, AV, AY, BB, BE, BH and BK);
- STW by material (obtained by the division of TW/total floor area_ Columns AK, AN, AQ, AT, AW, AZ, BC, BF, BI and BL);
- Percentage by material (obtained by the division of STEE by material/STEE of entire component_ Columns AL, AO, AR, AU, AX, BA, BD, BG, BJ and BM);
- Average STW by material (obtained by the average of the 10 buildings' STW_ Column BN).

*2.5. Database 04: STEE and STW Results of the Entire Building*

This database shows a summary of the ten buildings of each period. STEE and STW are organized by components by each house in the four periods. The shown values for P4 correspond to the STEE with (P4) and without (P4*) considering the industry improvements. These values are shown in the file named "04_STEE_STW_entire_Building", which is organized as follows:

- Period_House (Column A);
- Floor Area (Column B);
- Year of construction (Column C);
- STEE of Structure component (Column D);
- STEE of Envelope component (Column E);
- STEE of Finishes component (Column F);
- STTE of Entire Building (Column G);
- STW of Structure component (Column H);
- STW of Envelope component (Column I);
- STW of Finishes component (Column J);
- STW of entire building (Column K).

## 3. Methods

The process for obtaining the data shown in this article has divided into three parts: the Sample Selection, the Specific Embodied Energy Collection and the Calculations. The data collected focus on four periods: P1 (1980–1990), P2 (1991–2000, P3 (2001–2010) and P4 (2011–2020).

The selection of samples focuses on low-rise single-family buildings because it is the predominant typology in this country [41]. From every period, ten samples were analyzed to give a broader sample of the evolution of construction systems in this region. From a digital record of the Municipality of Cuenca [42], several sectors of the city were chosen where the main land use is residential. The chosen sectors were Misicata, Ricaurte and Miraflores, since they have construction from all periods mentioned before. Once the sectors were selected, a random pre-selection of buildings was carried out taking into account the construction period and the building typology. The first pre-selection shortlist included a total sample of 61 residential buildings. Although the municipality has a digital record of the buildings built, it only has informative data and no architectural or construction details. Due to this, it was proposed to carry out a field survey to collect the information on these buildings (architectonic plans, structural plans and technical specifications). The first step was to contact the homeowners and request access to this information. However, 21 buildings were discarded as the owners did not agree to provide the requested information. Therefore, the final sample was 40 buildings. Approved plans and technical specifications were obtained from the majority of this sample. However, the planimetric surveys of several houses had to be carried out in situ. This survey was accompanied by an interview with the owner to learn about the construction systems applied in addition to a detailed

inspection of the home. Once all this information was collected, a digital redraw was made to calculate the BoM of each house.

In parallel, the Specific Weight (SW) and the Specific Embodied Energy (SEE) value of each material found in the forty samples was established with the use of bibliography [13,23,26,27,35,36,38,43–52]. These values and their respective sources are specified in Database 01. In reference to SW, all values were obtained by bibliography from local industries. Nonetheless, most of the SEE values were obtained from other countries' databases due to the non-existence of an Embodied Energy inventory in Ecuador. Only four materials belong to Ecuadorian database. The selection of the SEE values focused, wherever possible, on a Cradle-to-Gate boundary condition. Nevertheless, 8 materials corresponded to Cradle-to-Site. The difference of these two boundary conditions is not representative since these eight materials have high SEE and high density [44]. In addition, three of the four materials belong to an Ecuadorian database referred as Gate-to-Gate. According to Alvear and Palomeque [27], since the extraction of raw materials was carried out by human beings and the quarry is located in the same place as the factory, the energy of these two processes is negligible. It is necessary to clarify that all the materials shown in this article have been assumed to have totally national production.

In addition to the boundary condition, another restriction to the SEE selection was the similarity of manufacturing processes, transportation and raw material origins with respect to Ecuadorian materials. Finally, since this article addresses the information from four different periods, industry improvements over these four decades have been considered. According to Kumar et al. [2], temporal changes have an important impact on SEE values. Based on an extensive investigation of the Ecuadorian construction industries, four materials have undergone a change referring to their manufacturing process (concrete, metal, aluminum and galvanized metal). These changes have only been consolidated in the fourth period, mainly due to the use of recycled materials and the implementation of sustainability considerations. As explained in the Data Description section, these four materials have two SEE values. The first one refers to material without considering the industry improvements applied to P1, P2 and P3. The second one refers to a product with the industry improvements consideration used only in P4.

The final stage, calculations, began with the quantification of the total volume (V) of every material from the BoM obtained from the 40 buildings collected in the Sample Selection stage. These calculations were organized into three components: Structure, Envelope and Finishes. Once V was obtained, the STW and STEE values of each material, component and of the entire building were quantified with the use of Equations (1) and (2):

$$STW = \sum(V \times SW)/S, \tag{1}$$

where STW is the Specific Total Weight ($kg/m^2$), V is the total volume of each material ($m^3$), SW is the Specific Weight of each material ($kg/m^3$) and S is the total floor, area which indicates the sum of the area of each floor measured to the outer surface of the exterior walls, excluding the front yard, back yard and garage.

$$STEE = \sum(STW \times SEE), \tag{2}$$

where STEE is the Specific Total Embodied Energy ($MJ/m^2$), STW is the Specific Total Weight of every material ($kg/m^2$) and SEE is the Specific Embodied Energy of every material ($MJ/kg$).

## 4. Graphical Representation: Total Results and Correlations

First of all, Figure 1 shows the STEE and STW values of the entire building made up of the values of the three components analyzed: Structure, Envelope and Finishes, from period 1 to period 4.

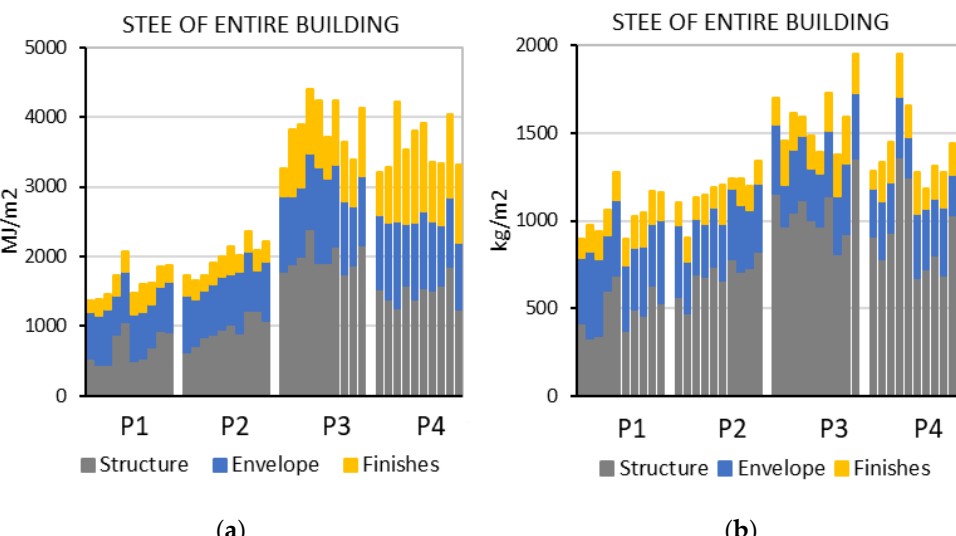

**Figure 1.** (**a**) STEE and (**b**) STW of the entire building and the values of each component, of the ten samples in each period.

According to this information, the STEE and STW of the entire building and its three components have undergone an increment in the last four decades. The STEE value of the entire building has increased 2.19 times, and STW by 1.36 times. The most representative change, in terms of STEE and STW, occurred between period 2 and period 3, which coincides with the adoption of dollarization in Ecuador, implemented from the year 2000.

It is important to note that the last period (P4) shows a reduction in both STEE and STW values with respect to period 3 (P3). In the case of the reduction of STEE, this is mainly due to the industrial improvements that concrete and metal have experienced in the last decade due to the sustainability regulations implemented from approximately 2010. Regarding the reduction of STW, this responds to the change from concrete to metal as the structure of the building.

Regarding the Structure, in its STEE and STW values this component experienced its most significant increase in period 3, due to the substitution of wood for concrete and metal in all structural elements. More importantly, the data from this region show that this component is the most representative of the four periods, both in terms of STEE and STW, as shown in other studies [16]. This can probably imply a close relationship between weight and embodied energy demand.

Due to the importance of the Structure, both in terms of STW and STEE, several previous studies have focused on reducing the environmental impact of the use of metal and mainly the use of concrete. As mentioned before, one of the most studied implementations has been the use of recycled aggregate concrete (RAC). Although the use of construction waste is a great strategy to reduce the environmental impact of the structure of buildings, this presents greater brittleness compared to natural aggregate concrete (NAC) [53,54] in its first stage and in its evolution over time [55]. The biggest problem concluded by several studies is that the use of recycled aggregates reduces the resistance of concrete [21,56,57]. Therefore, several solutions consider the addition of other materials to solve this problem. For example, the use of carbon, steel, and synthetic fibers to solve the dynamic strength problems has been addressed in the study by Selyutina and Smirnov [54]. Likewise, another study [21] analyzed the addition of steel fiber and poly-propylene to obtain a new green material: fiber-reinforced recycled aggregate concrete (FRAC).

In reference to the Envelope, although the STEE values in absolute terms have increased, the representativeness of this component with respect to the entire building has decreased. This same trend is shown in the values of STW. This mainly depends on the insertion of other materials (such as the hollow pumice block) and the increase in the glazed surface, which, although lighter, has a very high EE.

Finally, in the first period, the Finishes are the least representative component of the entire building both in terms of STEE and STW. However, this component shows the largest rate of increase among the three components. Even in period 4, the Finishes are more representative than the Envelope. This is due to the insertion of new materials such as stainless steel for handrails, porcelain tile for floors and mainly to the insertion of chipboard for doors, floors or fitted furniture.

According to the information shown in this article, the changes experienced by construction processes in their STEE and STW depend on the insertion of new materials such as concrete, metal or stainless steel. However, these changes depend to the same or a greater extent on the change in architectural models adopted in recent decades, such as the greater use of glass surfaces or flat roofs.

In addition, the information shown in Figure 1 highlights some correlations between different variables. Because of this, in the following subsections a regression analysis of STW, STEE and Year of Construction of the entire building and each component is performed.

### 4.1. Year of Construction and STEE Correlation

On the one hand, in this section the variable of the Year of Construction is related to the STEE values of the entire house (Figure 2). According to these values, the STEE of the entire house and the Year of Construction have a positive correlation, with an $R^2$ of 0.7324. These results indicate that buildings from the last decade tend to consume much more than buildings from the 80s. If this trend continues with the same slope, by 2050 a house will require an embodied energy of 6000 MJ/m$^2$.

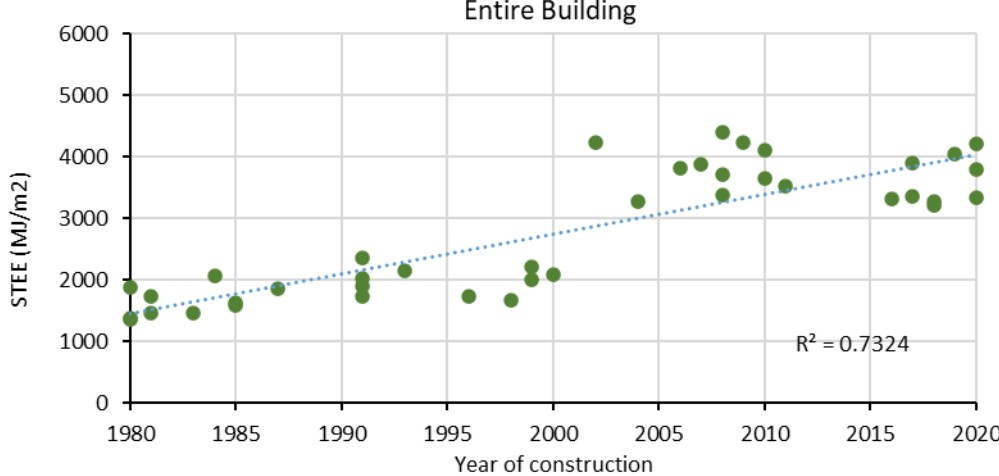

**Figure 2.** Correlation between the Year of Construction and STEE value of the entire building across the 40 samples.

On the other hand, the analysis of the influence of the Year of Construction on the STEE of each component is shown in Figure 3. These results show that the year of construction and the STEE have a positive correlation in the three components, with a coefficient of determination greater than 0.50. The highest correlation is for Structure, followed by Finishes. Therefore, these components are the ones that have had the greatest increase in the last four decades. This is related to the change in anti-seismic regulations and the changes in materials and architectural models that the region has experienced.

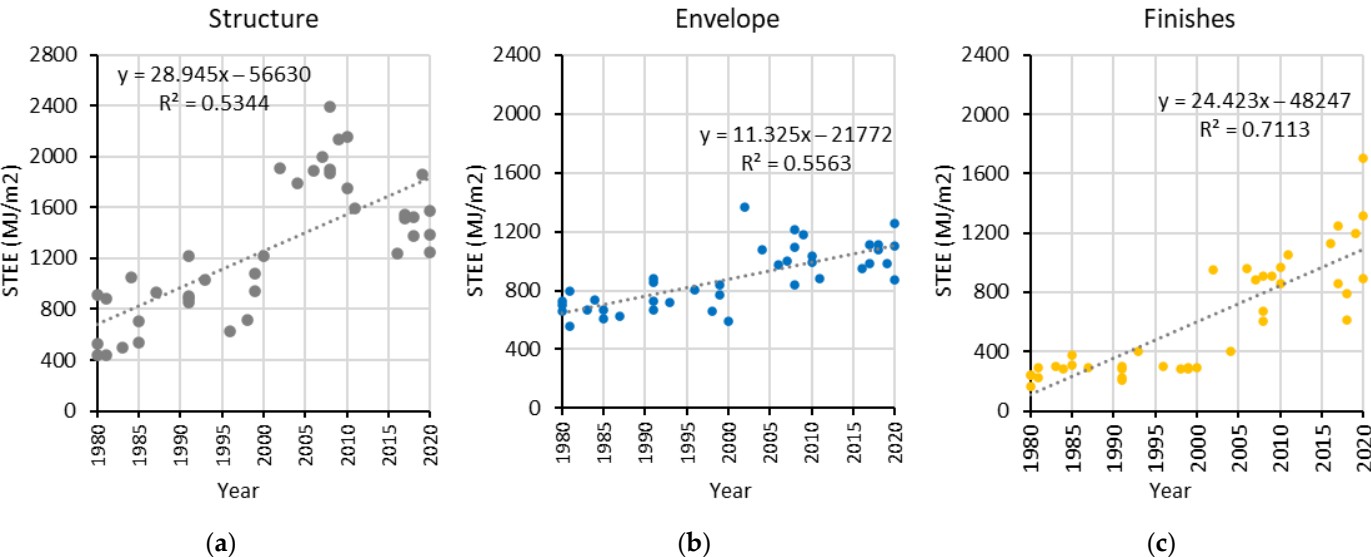

**Figure 3.** Correlation between Year of Construction and STEE value of (**a**) Structure, (**b**) Envelope and (**c**) Finishes components of the 40 samples.

*4.2. Year of Construction and STW Correlation*

First, the variable of the Year of Construction is related to the value of the STW of the entire building. Figure 4 shows that these two variables have a positive correlation, although with a slight slope and with an $R^2$ value of 0.4083. This indicates that the STW of the whole building does not depend as significantly on the Year of Construction as it does on STEE.

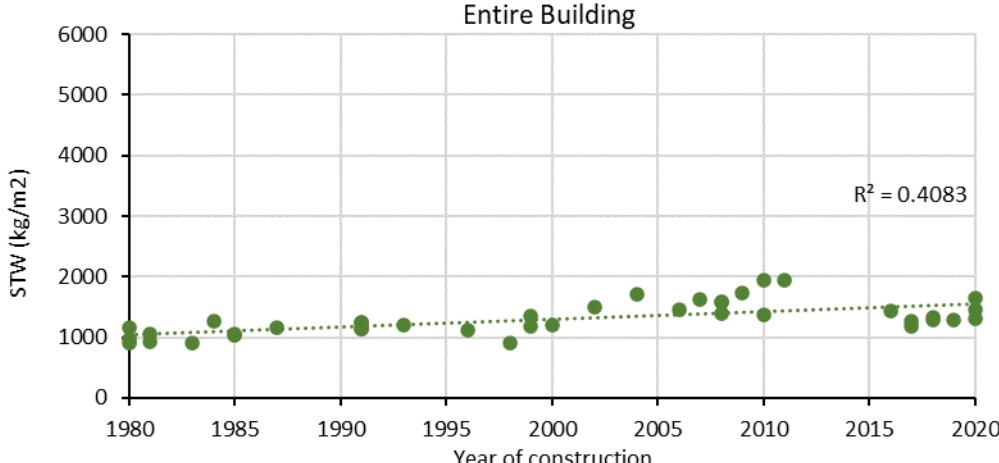

**Figure 4.** Correlation between Year of Construction and STW value of the entire building across the 40 samples.

Similarly, the STW values of each component do not present a high correlation with the Year of Construction (Figure 5). In the Structure and Finishes component, the correlation is positive, with $R^2$ values of 0.4716 and 0.1388, respectively, while in the case of the Envelope, the relationship is negative and its $R^2$ is 0.2702.

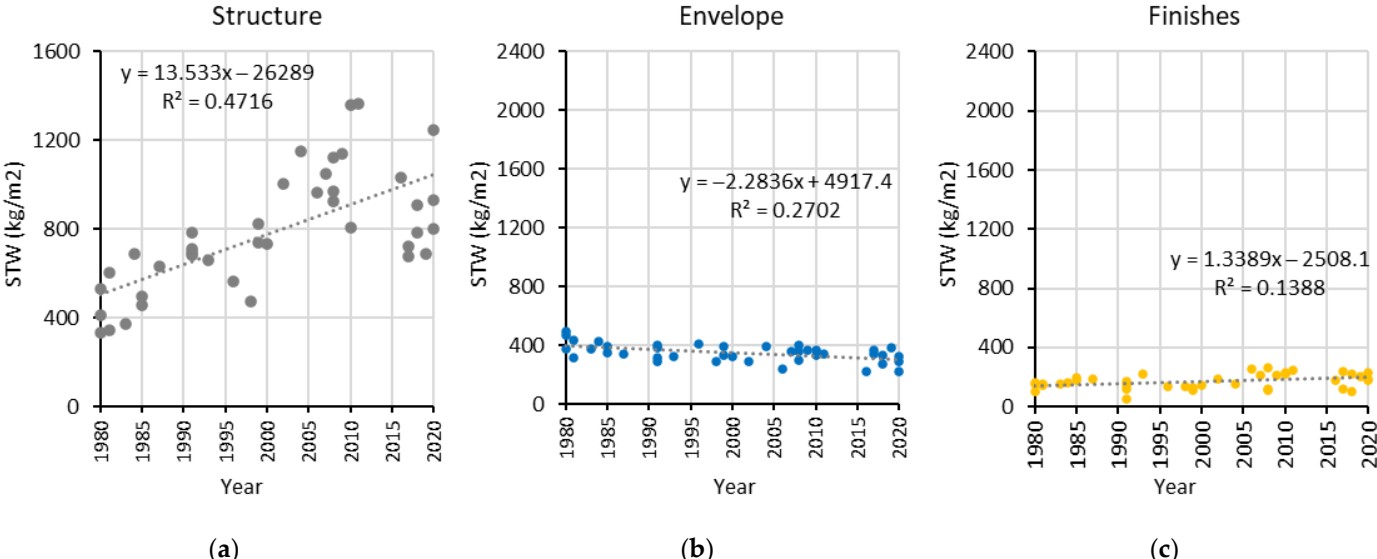

**Figure 5.** Correlation between Year of Construction and STW value of (**a**) Structure, (**b**) Envelope and (**c**) Finishes components of the 40 samples.

### 4.3. STEE and STW Correlation

The last analyzed correlation is that between STEE and STW. In the entire building (Figure 6), the regression analysis shows a high positive correlation between STEE and STW with an $R^2$ of 0.65. This shows that the total embodied energy demand of a building depends to a great extent on its weight.

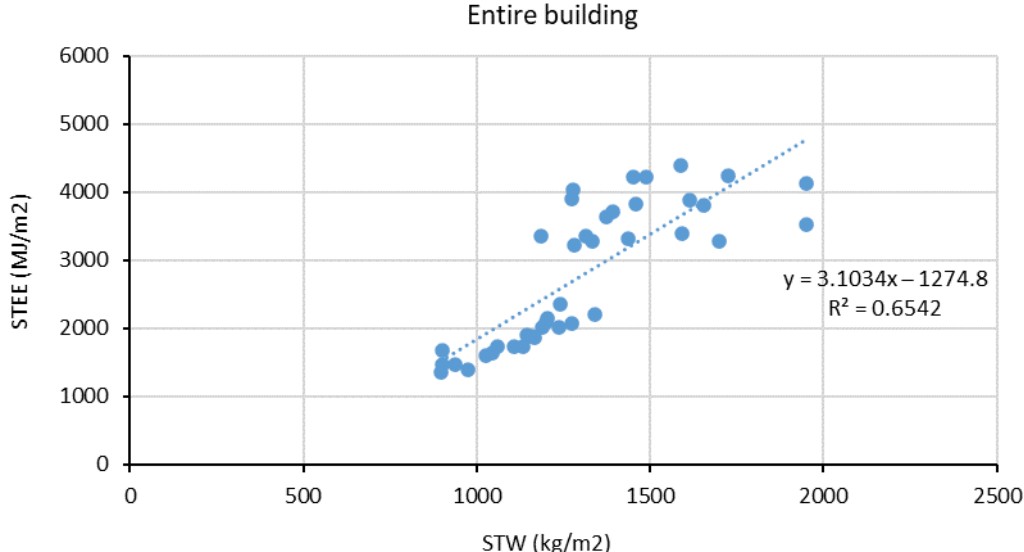

**Figure 6.** Correlation between STEE and STW values of the entire building across the 40 samples.

As in the previous sections, for a deeper analysis, these values (STEE and STW) have been correlated for each component. In the case of the Structure (Figure 7a), the $R^2$ value is 0.67, which is the highest positive correlation of the three components. According to this analysis, the value of STEE can be calculated as a function of STW with Equation (3):

$$STEE = 3.1034\,(STW) - 1274.8 \tag{3}$$

Regarding the Envelope component, the $R^2$ correlation is not significant and also has a negative trend. This supports the previously mentioned results, where the STEE value of this component increased even as the STW value decreased. Even though the correlation is not significant, a calculation model for the STEE of this component is presented in Equation (4):

$$STEE = -1.0912(STW) + 1265.1 \qquad (4)$$

The relationship of STEE and STW in the Finishes component has an $R^2$ of 0.3356 with a positive trend. Equation (5) shows the STEE calculation model as a function of STW. This equation highlights the fact that Finishes, even when they do not have a high weight value, tend to greatly increase the value of the incorporated energy.

$$STEE = 4.6687(STW) - 183.37 \qquad (5)$$

The results shown in this article are very relevant to show the trend of construction and architectural changes that this region has undergone in recent decades. However, it is important to point out some limitations of this study.

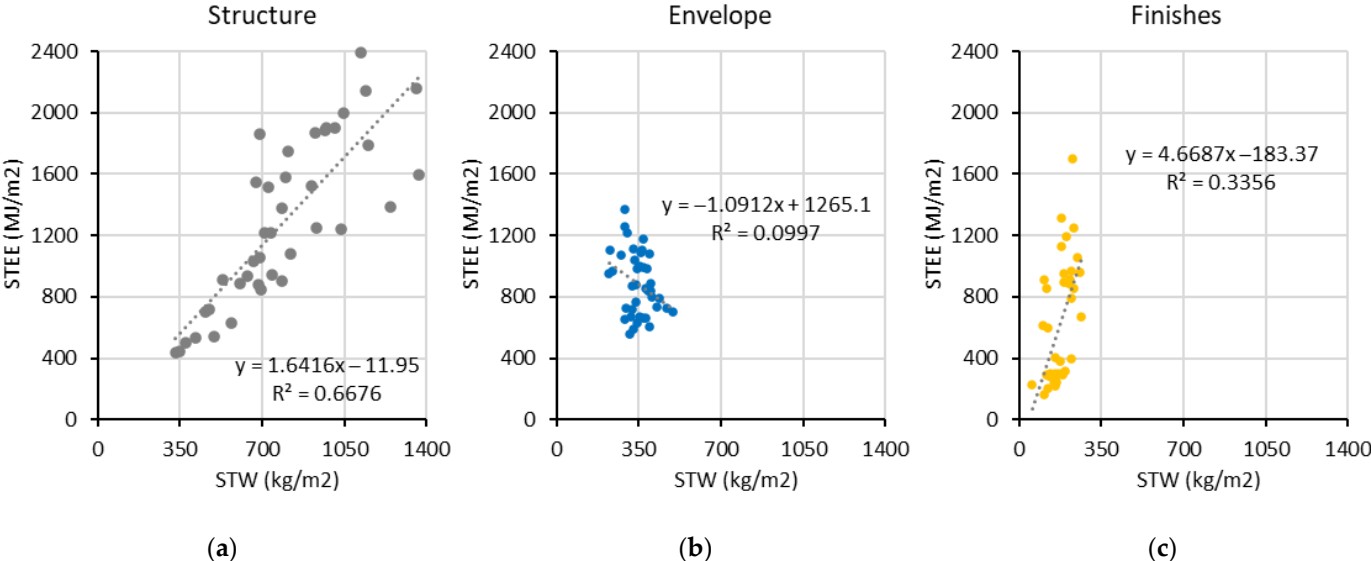

**Figure 7.** Correlation between STEE and STW values of (**a**) Structure, (**b**) Envelope and (**c**) Finishes components of the 40 samples.

The information shown for each sample addresses three components. However, it does not consider technical installations that can have a significant impact on the overall STEE value of the building.

Furthermore, although the results shown in this article take into account 40 residential buildings, it is important to note that a larger sample could improve the error margins of the displayed correlations. However, the greatest limitation of this study has been the collection of information from the samples. Both in the study region and in the country, the database held by the municipality is minimal and limited to general data. Therefore, no architectural or structural plans or BoMs exist for the residential buildings constructed in the city. Although the field survey made it possible to solve the lack of information, the inaccessibility of plans and the permission of the owners was a determining factor that reduced the number of samples initially planned (60).

## 5. Conclusions

The data presented in this article address the constructive evolution of the Andean region of Ecuador from the perspective of its embodied energy and its weight. The objective of these data was to show the changes that residential buildings have undergone in the

last four decades, focusing on their materials and the architectural decisions made in each period. Some general conclusions have been drawn from this information:

The change in construction systems in the last four decades has resulted in an increase in the STEE values of all components, and therefore of the entire building. The STEE and STW of the Structure component increased by 2.14 times and 1.92 times, respectively.

The Envelope's STEE was increased by 1.53 times, while its STW value was reduced by 21%. Lastly, Finishes was the component that had the greatest increase in terms of STEE, with an increase of 3.96 times, and its weight increased by 1.22 times. Consequently, the entire building had an increase in its STEE value of 2.19 times and in its STW value of 1.36 times.

In addition, the results show that the component that has the greatest impact on the total STEE of the building in all periods is the Structure. However, in the last period, the three components of Structure, Envelope and Finishes represent a similar value, with 41%, 29% and 30%, respectively. This implies that the relevance of the Structure in this region is no longer decisive compared to the other two components. Therefore, this study supports the conclusion that strategies to reduce the environmental impact of materials must equally address these three components of buildings.

Finally, the improvements in the materials industry reflect a reduction in the STEE value in the last period compared to period 3. However, this reduction is small and the trend is still increasing compared to the first periods. Therefore, technological improvements are not as significant as the changes in architectural and construction models shown in the last four decades.

**Author Contributions:** Conceptualization, J.T.-Q.; methodology, J.T.-Q.; formal analysis, J.T.-Q.; data curation, J.T.-Q. and T.S.-Q.; writing—original draft preparation, J.T.-Q. and T.S.-Q.; writing—review and editing, J.T.-Q. and T.S.-Q. All authors have read and agreed to the published version of the manuscript.

**Funding:** Work endorsed by the Catholic University of Cuenca, project code: DAMA-215543.

**Institutional Review Board Statement:** Not applicable.

**Informed Consent Statement:** Not applicable.

**Data Availability Statement:** The data presented in this study are openly available in Mendeley https://data.mendeley.com/datasets/z2pf5hzsss.

**Acknowledgments:** The authors thank all the residential building owners who provided the technical information for this process.

**Conflicts of Interest:** The authors declare no conflict of interest.

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
