# Peer review of "Dataset of Specific Total Embodied Energy and Specific Total Weight of 40 Buildings from the Last Four Decades in the Andean Region of Ecuador"

_data_

Round 1
Reviewer 1 Report
Comments for the proposed paper are given below.
1. As a first consideration, since the authors have published the dataset the article deals with, we point out that the data format (.xlsx) is not considered to be the most appropriate format. It is suggested to use the JSON format, which is the most common file format for "tree-like" data that potentially has multiple layers, like the proposed data.
2. The proposed dataset construction exercise is appreciable, but it would be useful to include some further analysis, such as correlations between the different inputs, a discussion on possible outliers, strategies for missing or incorrect data.
3. Although the data collected is interesting, the dataset is based on only 10 buildings. it is suggested to extend the representative samples considerably
4. it is strongly suggested to follow the FAIR (Findability, Accessibility, Interoperability, Reusability) principles for data, to improve the Findability, Accessibility, Interoperability, and Reuse of digital assets. The principles emphasise machine-actionability (i.e., the capacity of computational systems to find, access, interoperate, and reuse data with none or minimal human intervention).
Author Response
Please see the attachement

Reviewer 2 Report
This manuscript provides a report of the data set of specific total embodied energy and specific total weight of 40 buildings from the last four decades in the Andean region of Ecuador. Some suggestions for the revision of the paper are as follows:
1) A comma appears after the title, please delete it.
2) The abstract should be refined to show the primary conclusion of this study. In addition, the last sentence of the abstract is too ambiguous in its engineering significance and application.
3) Please highlight the research progress of structural embodied energy and specific total weight, as well as its research significance and engineering application value. The reader cannot see the practical significance of this study.
4) In the introduction section, the author should further highlight the shortage of the previous investigation and the innovation of this study, which is important for a scientific paper.
5) It is proposed in the paper that 25 kinds of materials have been investigated and studied, but no specific explanation has been made. Especially, green building materials are the direction of green development of civil engineering. Please supplement and discuss some properties of green building materials. The following references may be beneficial to this work: “Unloading and reloading stress-strain relationship of recycled aggregate concrete reinforced with steel/polypropylene fibers under uniaxial low-cycle loadings”. “Hysteresis and damping properties of steel and polypropylene fiber reinforced recycled aggregate concrete under uniaxial low-cycle loadings”
6) Some references cited are outdated, it is suggested to add the newly published papers.
7) A section in this paper on “Research Significance” is omitted.
8) The conclusions should be revised to outline the contribution of this study.
9) In this paper, the author only summarizes the existing materials and buildings and does not display the test database and conduct an in-depth analysis. How these works are carried out and why they are not shown in the article.
10) It gives readers the impression that the analysis and review of this work are not deep enough. Many existing formula models are put in the article, but the author lacks further analysis. On this basis, the author should propose new functional models through data regression analysis.
11) Only one figure is given in the article, and there is no in-depth analysis. The various information in Fig. 1 is not provided. the figures in this paper lack discussion and analysis. Please supplement and analyze.
Round 2
Reviewer 1 Report
the answers provided fully respond to the comments made.
I consider that the article can be published as is.
Reviewer 2 Report
The authors have addressed all the comments, and this manuscipt is suggested to be accpeted as is.